# Fuzzy Time-Varying Formation Control for Unmanned Surface Vehicles Considering Aerial Base Station Allocation Algorithm

Qihe Shan
*Navigation College*
*Dalian Maritime University*
Dalian, China
shanqihe@163.com

Peiyun Ye
*Navigation College*
*Dalian Maritime University*
Dalian, China
peiyunye2019@gmail.com

Fei Teng*
*College of Marine Electrical Engineering*
*Dalian Maritime University*
Dalian, China
brenda_teng@163.com

Tieshan Li
*School of Automation Engineering*
*University of Electronic Science and Technology of China*
Chengdu, China
tieshanli@126.com

Qi Xu
*Research Institute of Intelligent Networks*
*Zhejiang Lab*
Hangzhou, China
xuqi@zhejianglab.com

*Abstract*—As an essential maritime relay station, the allocation of a limited number of Aerial Base Stations (ABSs) to meet the communication needs of unmanned surface equipments has emerged a vital issue. This paper investigates the selection of formation centers for time-varying formation control of nonlinear Unmanned Surface Vehicles (USVs) assisted by ABSs. First, considering the Laplacian matrix among the leader ABSs and follower USVs, a communication volume-based and position-dependent algorithm is developed to determine the assigned ABSs for each USV. Next, leveraging fuzzy approximators, a fuzzy time-varying formation control protocol for the USVs is proposed, with the assigned ABS serving as the formation center. Furthermore, based on Lyapunov stability theory, it is demonstrated that all USVs successfully achieve the predefined time-varying formation structure while rotating around the selected ABS. Finally, a numerical simulation is provided to validate the effectiveness of the proposed theoretical results.

*Index Terms*—aerial base stations allocation, Laplacian matrix, unmanned surface vehicles, time-varying formation, fuzzy approximation

## I. Introduction

With the introduction of an innovative network architecture supporting polymorphic domain name identification, including content, identity, IP address, and geospatial location [1], it has become feasible to establish communication networks for Aerial Base Stations (ABSs) that support multiple offshore unmanned systems. ABSs, functioning as relays between terrestrial base stations and the offshore unmanned equipment, have attracted significant attention regarding their applications. Depending on the devices being supported, ABSs can be deployed in low Earth orbit, high-altitude, or low-altitude environments [2]. For offshore unmanned devices, Unmanned Surface Vehicles (USVs), serving as the primary force in maritime operations, are employed in a wide range of applications, including marine search and rescue [3], ocean mineral exploration [4], and water monitoring [5]. However, harsh maritime environments or sensor failures may disrupt communication within USV systems, at which point ABSs serve as edge computing units can provide computational power and communication support. Therefore, it is preferable for some USV systems to operate near the ABSs.

An interesting research question remains in determining how a limited number of ABSs can effectively support numerous number of USVs while ensuring they are evenly distributed around their designated ABSs. Among these, in relation to ABSs' deployment problem, most of the existing literature has explored optimization algorithms for determining the ABSs' deployment based on power allocation, data traffic demand, throughput, and reducing transmission delay [6]. Nonetheless, these approaches often result in long computation times. Communication topology, which indicates whether agents can communicate with each other, can efficiently characterize the communication volume between agents through simple calculations [7]. If the system's Laplacian matrix is available and the number of ABSs and maritime unmanned devices is known, the ABS allocation problem, based on communication volume and position, can be solved more efficiently using containment control theory than with traditional optimal methods.

A group od USV systems performing time-varying formation tasks around a designated ABS can facilitate the even distribution of USVs. Over the past two decades, the challenges

This work was supported in part by the National Natural Science Foundation of China (grant numbers 51939001, 61976033, 52371360, 52201407, 61751202), in part by the Liaoning Revitalization Talents Program (grant number XLYC1908018), in part by the Fundamental Research Funds for Central Universities (Grant number 3132024118), and in part by the Zhejiang Lab Open Research Project (grant number K2022QA0AB03). (Corresponding author: Fei Teng).

of formation control for USVs, aimed at providing comprehensive area coverage, have garnered significant interest. In [8], a rigid formation control for scaling the single integrator USV model was investigated. To accommodate more complex formation objectives and more accurate models, a distributed guidance law based on a path maneuvering algorithm was proposed in [9] to achieve the desired time-varying formation. Additionally, an extended state observer was introduced to compensate for the effects of uncertainties in the USV model. Due to the superior approximation capabilities, intelligent adaptive algorithms have been extensively studied by many researchers. In [10], a leader-follower formation control approach using the backstepping procedure was developed for USV systems, where neural network approximators were employed to estimate the nonlinear dynamics. Notwithstanding, the collaborative frameworks between ABSs and USVs, along with more complex formation targets such as the selection of formation centers under the condition of limited base station equipment, have not been addressed in references [8–10], making this a valuable topic for further research.

Motivated by the aforementioned observations, this article proposes an ABS allocation algorithm that selects formation centers to enable the uniform distribution of USVs, effectively addressing the challenge of time-varying formation control with limited base station resources. The main contributions are twofold. First, regions formed by leader ABSs are evenly divided, and based on the information from the Laplacian matrix, the convex states of leader ABSs' positions are used to calculate the formation centers for each USV. Second, a distributed fuzzy formation control protocol is designed for follower USVs, where the nonlinearities are compensated by fuzzy approximators. Lastly, using Lyapunov stability theory, it is demonstrated that each USV can achieve a time-varying formation and rotate around its assigned ABS.

The rest of this paper is structured as follows. Preliminaries and model introductions are presented in Section II. Section III demonstrates the allocation algorithm and the stability analysis of the time-varying formation control method. Simulation examples are given in Section IV. Section V concludes this paper.

## II. PRELIMINARIES AND PROBLEM FORMULATION

### A. Preliminaries

*1) Graph theory:* The communication topology among $M$ leader ABSs and $N$ follower USVs is represented by an undirected graph $\mathcal{G}(\Psi, \mathcal{E}, \mathcal{W})$, where $\Psi = \{\psi_1, \psi_2, ..., \psi_{(M+N)}\}$, $\mathcal{E}$, and $\mathcal{W} = [w_{ij}] \in \mathbb{R}^{(M+N)\times(M+N)}$ are the node set, the edge set and the adjacency matrix, respectively. The agent i is considered a neighbor of the agent j. If and only if $(\psi_i, \psi_j) \in \mathcal{E}$, $w_{ij} > 0$. The Laplacian matrix is denoted as $\mathcal{L} = \mathcal{D} - \mathcal{W} \in \mathbb{R}^{(M+N)\times(M+N)}$, where $\mathcal{D} = \text{diag}\left\{\sum_{j=1}^{M+N} w_{1j}, \cdots, \sum_{j=1}^{M+N} w_{(M+N)j}\right\}$ represents the degree matrix. The graph $\mathcal{G}$ is assumed to be connected. Let $\mathcal{I}_L$ and $\mathcal{I}_F$ be the leader and the follower sets, respectively.

The Laplacian matrix can be expressed in the following form:

$$\mathcal{L} = \begin{bmatrix} \mathbf{0}_{M\times M} & \mathbf{0}_{M\times N} \\ -\mathcal{L}_1 & \mathcal{L}_2 \end{bmatrix} \in \mathbb{R}^{(M+N)\times(M+N)},$$

where $\mathcal{L}_1 \in \mathbb{R}^{N\times M}$ and $\mathcal{L}_2 \in \mathbb{R}^{N\times N}$.

**Lemma 1** *[11] All eigenvalues of $\mathcal{L}_2$ are positive values. Each row of matrix $\mathcal{L}_2^{-1}\mathcal{L}_1$ sums to one, and all its elements are nonnegative.*

*2) Fuzzy approximation [12]:* For any given smooth continuous function $G(\xi)$ defined on a compact set $\Upsilon \in \mathbb{R}^n$ and any constant $\tau > 0$, there exists a fuzzy logic system $W^T\varphi(\xi)$ such that

$$\sup_{\xi\in\Upsilon} \left|G(\xi) - W^T\varphi(\xi)\right| \le \tau, \tag{1}$$

where $W$ is a fuzzy parameter vector and $\varphi(\xi)$ is a fuzzy basis function. The optimal parameter vector $W^*$ can be defined as

$$W^* = \arg\min\left[\sup_{\xi\in\Upsilon}\left|G(\xi) - W^T\varphi(\xi)\right|\right], \tag{2}$$

then, $G(\xi)$ can be expressed as

$$G(\xi) = W^{*T}\varphi(\xi) + \tau. \tag{3}$$

### B. Model description

For the $k$th leader ABS, which can be modeled as an Unmanned Aerial Vehicle (UAV), due to its high mobility, cost-effectiveness, and flexible deployment capabilities [13]. The dynamics model of $k$th UAV is given as follows:

$$\dot{P}_k = AP_k + Bu_k, k \in \mathcal{I}_L, \tag{4}$$

where $P_k = [x_k, y_k, z_k, v_{xk}, v_{yk}, v_{zk}]^T \in \mathbb{R}^6$ is the state vector combines positions and velocities, $u_k \in \mathbb{R}^3$ is the control input of the UAV. $A = [\mathbf{O}_{3\times3}, \boldsymbol{I}_3; \mathbf{O}_{3\times3}, \mathbf{O}_{3\times3}]$. $B = [\mathbf{O}_{3\times3}, \boldsymbol{I}_3]^T$, where $\mathbf{O}_{3\times3}$ and $\boldsymbol{I}_3$ denote the $3 \times 3$ zero matrix and the identity matrix, respectively. The matrix pair $(A, B)$ is stabilizable.

The kinematics and dynamics model of the $i$th follower USV [14] can be expressed as follows:

$$\dot{P}_i = AP_i + B(f_i + u_i), i \in \mathcal{I}_F, \tag{5}$$

where $P_i = [\eta_{xi}, \eta_{yi}, \phi_i, \mu_i]^T \in \mathbb{R}^6$ is the positions and velocities integrated vector, where $\mu_i = R(\phi_i)v_i \in \mathbb{R}^3$, and $R(\phi_i) \in \mathbb{R}^{3\times3}$ is the rotation matrix. $v_i = [v_{xi}, v_{yi}, v_{\phi i}]^T \in \mathbb{R}^3$ is the velocity vector. $f_i \in \mathbb{R}^3$ is the model inner nonlinear term which expressed as $f_i = J_i\mu_i - R^T(\phi_i)M_i^{-1}(C_i + D_i)R^T(\phi_i)\mu_i$. $J_i = [0, -v_{\phi i}, 0; v_{\phi i}, 0, 0; 0, 0, 0] \in \mathbb{R}^{3\times3}$. $C_i \in \mathbb{R}^{3\times3}$, $M_i \in \mathbb{R}^{3\times3}$ and $D_i \in \mathbb{R}^{3\times3}$ denote the Coriolis matrix, the inertia matrix, and the damping matrix, respectively. $u_i \in \mathbb{R}^3$ is the control input. More detailed definitions of the parameters for the USV model can be found in [14].

### C. Control objective

In this article, all leader ABSs are assigned to the corresponding USVs based on the Laplacian matrix information firstly. Subsequently, follower USVs are designed to achieve the desired time-varying formation, with the assigned ABS acting as the formation center.

The time-varying formation structure for the $i$th USV is specified by $h_i(t) = [h_{ix}, h_{iy}, h_{i\phi}, \dot{h}_{ix}, \dot{h}_{iy}, \dot{h}_{i\phi},] \in \mathbb{R}^6$, where the function $h_i$ is expected to be differentiable.

**Assumption 1** *All ABSs fly at the same altitude, and the geometric arrangement of their position coordinates forms a regular polygon.*

**Remark 1** *This algorithm currently considers only the scenario where the horizontal projections of the ABS positions form a regular polygon. In practical applications, the ABS heights may vary, but as long as their horizontal projections form a regular polygon, the algorithm remains applicable. For simplicity, Assumption 1 imposes specific constraints on both the height and position of the ABS. The case of varying ABS heights and arbitrary positions will be addressed in future research.*

**Definition 1** *For any $P_i(0)$ is bounded and $\vartheta > 0$, the $i$th follower USV is said to achieve the time-varying formation, if the following equation is satisfied that*

$$\lim_{t \to \infty} \|P_i - h_i - P_{\eta_i}\| \leqslant \vartheta, i \in \mathcal{I}_F, \tag{6}$$

*where $P_{\eta_i} \in \mathbb{R}^6$ represents the formation center state vector corresponding to the allocated ABS. $\vartheta$ is called the time-varying formation error bound.*

## III. MAIN RESULTS

In this section, an ABS allocation method based on the Laplacian matrix for ABS-USV systems is first devised. The algorithm outputs the allocation order of ABSs for each USV based solely on the input of the matrix $\mathcal{L}_2^{-1}\mathcal{L}_1$ and the positions of ABSs. Next, a fuzzy time-varying formation control protocol is proposed, with its stability proven in the final step.

### A. Laplacian matrix-based ABS allocation algorithm

**Step 1.** Generate the coordinates. Suppose Assumption 1 holds, thus the position for the $k$th leader ABS can be expressed as follows:

$$\begin{cases} x_k = r\cos[(k-1)\frac{2\pi}{M}], \\ y_k = r\sin[(k-1)\frac{2\pi}{M}], k \in \mathcal{I}_L, \end{cases} \tag{7}$$

where $r$ denotes the radius of the regular polygon.

**Step 2.** Regions division. In light of (7), the midpoint of the $k$th edge for the regular polygon can be obtained as

$$\begin{cases} x_{ka} = x_k + \dfrac{x_{k+1} - x_k}{2}, k+1 \neq M, \\ y_{ka} = y_k + \dfrac{y_{k+1} - y_k}{2}, k+1 \neq M, \\ x_{k+1} = x_1, k+1 = M, \\ y_{k+1} = y_1, k+1 = M. \end{cases} \tag{8}$$

Then, connect the $k$th midpoint $H_k$ with the $(M-M/2+k)$th midpoint $(H_{M-M/2+k})$. The perpendicular bisectors of the $k$th edge and $(M-M/2+k)$th edge can be represented by the line segment $(H_k, H_{M-M/2+k})$. Finally, the regular polygon can be divided into $M$ equal-area regions by $M$ perpendicular bisectors.

**Step 3.** Calculate the included angle. Let $\sum_{k=1}^{M} \beta_{ik} P_k = P_{ci} \in \mathbb{R}^6$ be the convex combination states of leader ABSs, where $\beta_{ik}$ is the element of the matrix $\mathcal{L}_2^{-1}\mathcal{L}_1$. Then, the angle between the $k$th convex combination state and the surge forward direction can be calculated as

$$\begin{cases} \theta_{ci} = atan(y_{ci}, x_{ci}), \theta_{ci} > 0, \\ \theta_{ci} = \theta_{ci} + 2\pi, \theta_{ci} < 0, \end{cases} \tag{9}$$

where $x_{ci}$ and $y_{ci} \in P_{ci}$.

**Step 4.** ABSs allocation. The index of the nearest ABS to each convex state can be obtained as

$$\begin{cases} \eta_i = floor(\dfrac{\theta_{ci} + 2\pi/M}{2\pi/M}) + 1, \eta_i < M+1, \\ \eta_i = 1, \eta_i \geq M+1, \end{cases} \tag{10}$$

where $floor(\cdot)$ is the function utilized to return the largest integer less than or equal to the given parameter. Thus, define $P_{\eta_i}$ as the formation center for the $i$th USV.

The diagram for the Laplacian matrix-based ABS allocation algorithm is summarized in Fig. 1, where the convex state $(x_{ci}, y_{ci})$ falls within region 5, indicating that the 5th ABS is allocated to the $i$th USV.

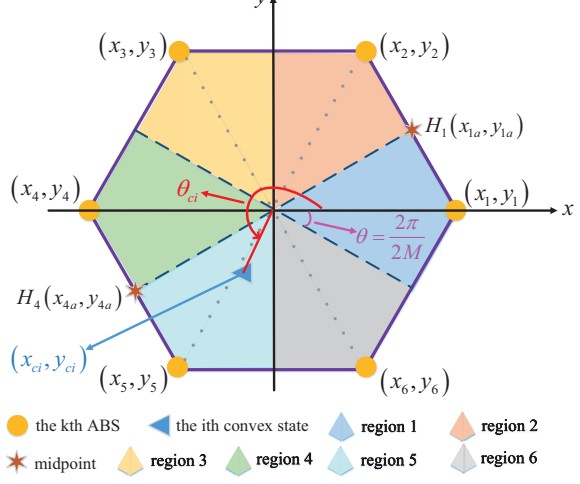

Fig. 1: The diagram for the region allocation.

**Remark 2** *In Section III-A, the Laplacian matrix and position-based allocation algorithm determines the assigned ABS index for each USV, enabling efficient base station distribution. In Section III-B, the control protocol drives each USV to its assigned ABS position, where each USV forms a formation centered on the ABS's horizontal position, achieving evenly distribution of USVs.*

### B. Time-varying formation control design

In this part, a time-varying formation protocol is developed for the follower USV to shape a formation structure and rotate around the selected ABS from Step 4.

Define the formation error $e = [e_{M+1}^T, e_{M+2}^T, \cdots, e_{M+N}^T]^T \in \mathbb{R}^{6N}$, in which $e_i = \sum_{j=M+1}^{M+N} w_{ij}[(P_i - h_i - P_{\eta_i}) - (P_j - h_j - P_{\eta_j})] + \sum_{k=1}^{M} w_{ik}(P_i - h_i - P_{\eta_i})(i \in \mathcal{I}_F, k \in \mathcal{I}_L)$. $w_{ik} > 0$ if the $i$th USVs has a connection with the $k$th ABS.

In accordance with the fuzzy approximation theorem, the nonlinear term $f_i$ in (5) can be expressed as follows:

$$f_i = W_i^{*T}\varphi_i(P_i) + \tau_i, \qquad (11)$$

where $W_i^* \in \mathbb{R}^{q\times 3}$ is the optimal fuzzy parameter matrix, $\varphi_i(P_i) \in \mathbb{R}^q$ is the fuzzy basis vector, and $\tau_i \in \mathbb{R}^3$ is the approximate error.

Then, the formation protocol together with the fuzzy update laws are designed as

$$u_i = \gamma K_1 e_i + K_2 h_i - \hat{W}_i^T\varphi_i(P_i), \qquad (12a)$$

$$\dot{\hat{W}}_i = \varpi_i\left[\varphi_i(P_i)e_i^T\mathcal{H}^{-1}B - \rho_i\hat{W}_i\right], \qquad (12b)$$

where $\gamma > 0, \varpi_i > 0$ and $\rho_i > 0$ are designed parameters. $\hat{W}_i$ is the estimate of $W_i$. $K_1 = -B^T\mathcal{H}^{-1} \in \mathbb{R}^{3\times 6}$. $\mathcal{H} \in \mathbb{R}^{6\times 6}$ is a positive definite matrix will be defined afterwards. $K_2 \in \mathbb{R}^{3\times 6}$ satisfies

$$\lim_{t\to\infty}[\dot{h}_i - (A + BK_2)h_i] = 0. \qquad (13)$$

Define $\zeta_i = P_i - h_i - P_{\eta_i}$, then, the errors and states can be written in following compact forms: $\zeta = [\zeta_{M+1}^T, \zeta_{M+2}^T, \ldots, \zeta_{M+N}^T]^T \in \mathbb{R}^{6N\times 1}$, $P_l = [P_1^T, P_2^T, \ldots, P_M^T]^T \in \mathbb{R}^{6M\times 1}$, $P_f = [P_{M+1}^T, P_{M+2}^T, \ldots, P_{M+N}^T]^T \in \mathbb{R}^{6N\times 1}$, $h = [h_{M+1}^T, h_{M+2}^T, \cdots, h_{M+N}^T]^T \in \mathbb{R}^{6N\times 1}$. $W = \text{diag}(W_{M+1}, W_{M+2}, \cdots, W_{M+N}) \in \mathbb{R}^{qN\times 3N}$, $\varphi(P_f) = [\varphi_{M+1}^T(P_{M+1}), \varphi_{M+2}^T(P_{M+2}), \cdots, \varphi_{M+N}^T(P_{M+N})]^T \in \mathbb{R}^{qN\times 1}$, $\tau = [\tau_{M+1}^T, \tau_{M+2}^T, \cdots, \tau_{M+N}^T]^T \in \mathbb{R}^{3N\times 1}$.

According to (2), (3) and $e = (\mathcal{L}_2 \otimes \mathbf{I}_6)\zeta$, one can obtained

$$\dot{\zeta} = [\mathbf{I}_N \otimes (A+BK_2)]h - \dot{h} + (\mathbf{I}_N \otimes A + \gamma\mathcal{L}_2 \otimes BK_1)\zeta - (\mathbf{I}_N \otimes B)\left[\tilde{W}^T\varphi(P_f) - \tau\right]. \qquad (14)$$

where $\tilde{W} = \hat{W} - W^*$.

**Theorem 1** *Under Assumption 1, if condition (13) hold, $\gamma \geqslant (\alpha_1/\sigma_{M+1} + 1/c_3)/2$, $\alpha_1 > 0$, $\alpha_2 > 0$, and the subsequent LMI condition satisfies:*

$$A\mathcal{H} + \mathcal{H}A^T - \alpha_1 BB^T + \alpha_2\mathcal{H} \prec 0, \qquad (15)$$

*where the positive definite matrix $\mathcal{H} \succ 0$ is the solution of (15), the fuzzy time-varying formation protocol (12a) and the parameter updated law (12b) designed for each follower USV achieve the predefined time-varying formation target, i.e. $\lim_{t\to\infty}\|P_i - h_i - P_{\eta_i}\| \leqslant \vartheta, i \in \mathcal{I}_F$.*

**Proof.** Construct the following Lyapunov functional:

$$V = \zeta^T(\mathcal{L}_2 \otimes \mathcal{H}^{-1})\zeta + \sum_{i=M+1}^{M+N}\text{tr}\left(\frac{1}{\varpi_i}\tilde{W}_i^T\tilde{W}_i\right). \qquad (16)$$

Taking the time derivative of $V$ along (12b) and (14), we can obtain

$$\dot{V} = 2\zeta^T(\mathcal{L}_2 \otimes \mathcal{H}^{-1}A - \gamma\mathcal{L}_2\mathcal{L}_2 \otimes \mathcal{H}^{-1}BB^T\mathcal{H}^{-1})\zeta - 2\zeta^T(\mathcal{L}_2 \otimes \mathcal{H}^{-1}B)\left[\tilde{W}^T\varphi(P_f) - \tau\right] + 2\zeta^T(\mathcal{L}_2 \otimes \mathcal{H}^{-1})\bar{h} + 2\sum_{i=M+1}^{M+N}\text{tr}\left[\tilde{W}_i^T\varphi_i(P_i)e_i^T\mathcal{H}^{-1}B - \rho_i\tilde{W}_i^T\hat{W}_i\right], \qquad (17)$$

where $\bar{h} = [\mathbf{I}_N \otimes (A+BK_2)]h - \dot{h}$.

In view of Young's inequality, one has

$$2\zeta^T(\mathcal{L}_2 \otimes \mathcal{H}^{-1})\bar{h} \leqslant c_1\zeta^T(\mathcal{L}_2 \otimes \mathcal{H}^{-1})\zeta + \frac{\sigma_{max}(\mathcal{L}_2 \otimes \mathcal{H}^{-1})\|\bar{h}\|^2}{c_1},$$

$$-2\sum_{i=M+1}^{M+N}\text{tr}\left(\rho_i\tilde{W}_i^T\hat{W}_i\right) \leqslant -\left(2 - \frac{1}{c_2}\right)\sum_{i=M+1}^{M+N}\text{tr}\left(\rho_i\tilde{W}_i^T\tilde{W}_i\right) + c_2\sum_{i=M+1}^{M+N}\text{tr}(\rho_i W_i^{*T}W_i^*), \qquad (18)$$

where $c_1 > 0, c_2 > 0$, and $\sigma_{max}(\cdot)$ ($\sigma_{min}(\cdot)$) stands for the maximum (minimum) eigenvalue of the relevant matrix.

Since the following equation holds

$$-2\zeta^T(\mathcal{L}_2 \otimes \mathcal{H}^{-1}B)\tilde{W}^T\varphi(P_f) = 2\sum_{i=M+1}^{M+N}\text{tr}\left[\tilde{W}_i^T\varphi_i(P_i)e_i^T\mathcal{H}^{-1}B\right], \qquad (19)$$

let $\bar{\zeta} = (\mathbf{I}_N \otimes \mathcal{H}^{-1})\zeta$, then, (17) can be further expressed as

$$\dot{V} \leq 2\bar{\zeta}^T(\mathcal{L}_2 \otimes A\mathcal{H} - \gamma\mathcal{L}_2\mathcal{L}_2 \otimes BB^T)\bar{\zeta} + 2\bar{\zeta}^T(\mathcal{L}_2 \otimes B)\tau - \left(2 - \frac{1}{c_2}\right)\sum_{i=M+1}^{M+N}\text{tr}\left(\rho_i\tilde{W}_i^T\tilde{W}_i\right) + c_2\sum_{i=M+1}^{M+N}\text{tr}(\rho_i W_i^{*T}W_i^*) + \frac{\sigma_{max}(\mathcal{L}_2 \otimes \mathcal{H}^{-1})\|\bar{h}\|^2}{c_1} + c_1\zeta^T(\mathcal{L}_2 \otimes \mathcal{H}^{-1})\zeta. \qquad (20)$$

Define a unitary matrix $Y = [\mathbf{1}_N/\sqrt{N}, U] \in \mathbb{R}^{N\times N}$ with $U \in \mathbb{R}^{N\times(N-1)}$, such that $Y^T\mathcal{L}_2Y \triangleq \text{diag}(\sigma_{M+1}, \sigma_{M+2}, \ldots, \sigma_{M+N}) \in \mathbb{R}^{N\times N}$, where $0 < Re(\sigma_{M+1}) \leq Re(\sigma_{M+2}) \leq \cdots \leq Re(\sigma_{M+N})$. Let $\hat{\zeta} = (Y^T \otimes \mathbf{I}_6)\bar{\zeta}$. If $\gamma$ can be appropriately chosen such that $\gamma \geqslant (\alpha_1/\sigma_{M+1} + 1/c_3)/2$, and

$2\bar{\zeta}^T(\mathcal{L}_2 \otimes B)\tau \leq \bar{\zeta}^T(\mathcal{L}_2\mathcal{L}_2 \otimes BB^T)\bar{\zeta}/c_3 + c_3\tau_m^2$, with $\tau_m = \sum_{i=M+1}^{M+N}\tau_i$, it can be deduced from (20) that

$$
\begin{aligned}
&\bar{\zeta}^T(\mathcal{L}_2 \otimes (A\mathcal{H} + \mathcal{H}A^T) - (2\gamma - 1/c_3)\mathcal{L}_2\mathcal{L}_2 \otimes BB^T)\bar{\zeta} \\
&= \sum_{i=M+1}^{M+N} \sigma_i \hat{\zeta}_i^T[A\mathcal{H} + \mathcal{H}A^T - (2\gamma - 1/c_3)\sigma_i BB^T]\hat{\zeta}_i \\
&< -\alpha_2 \zeta^T\left(\mathcal{L}_2 \otimes \mathcal{H}^{-1}\right)\zeta.
\end{aligned}
\tag{21}
$$

It can be further derived as

$$
\begin{aligned}
\dot{V} &< -(\alpha_2 - c_1)\zeta^T\left(\mathcal{L}_2 \otimes \mathcal{H}^{-1}\right)\zeta + \sigma_{max}\left(\mathcal{L}_2 \otimes \mathcal{H}^{-1}\right)\|\bar{h}\|^2/c_1 \\
&+ c_2 \sum_{i=M+1}^{M+N} \text{tr}(\rho_i W_i^{*T}W_i^*) - \left(2 - \frac{1}{c_2}\right)\sum_{i=M+1}^{M+N} \text{tr}\left(\rho_i \tilde{W}_i^T \tilde{W}_i\right) + c_3\tau_m^2 \\
&\leq -dV + \Omega,
\end{aligned}
\tag{22}
$$

where $d = \min\left\{\dfrac{(\alpha_2 - c_1)\sigma_{max}(\mathcal{L}_2 \otimes \mathcal{H}^{-1})}{\sigma_{min}(\mathcal{L}_2 \otimes \mathcal{H}^{-1})}, \left(2 - \dfrac{1}{c_2}\right)\sum_{i=M+1}^{M+N}\rho_i\right.$ $\left./\sum_{i=M+1}^{M+N}\dfrac{1}{\varpi_i}\right\}$, $\Omega = \sigma_{max}(\mathcal{L}_2 \otimes \mathcal{H}^{-1})\|\bar{h}\|^2/c_1 + c_2\sum_{i=M+1}^{M+N}\text{tr}(\rho_i W_i^{*T}W_i^*) + c_3\tau_m^2$. From (13) which leads to $\lim_{t\to\infty}\bar{h} = 0$. Then, by the Gronwall inequality [15], it is concluded that

$$
\lim_{t\to\infty}\|\zeta_i\| \leq \lim_{t\to\infty}\|\zeta\| = \sqrt{\frac{\bar{\Omega}}{d\sigma_{min}(\mathcal{L}_2 \otimes \mathcal{H}^{-1})}} = \vartheta, \tag{23}
$$

where $\bar{\Omega} = c_2\sum_{i=M+1}^{M+N}\text{tr}(\rho_i W_i^{*T}W_i^*) + c_3\tau_m^2$. Therefore, it can be deduced that the time-varying formation of the considered (4) and (5) systems under the ABS allocation algorithm is guaranteed by applying the presented control protocol. This completes the proof of Theorem 1. ∎

## IV. SIMULATION RESULTS

In this section, an illustrative simulation is presented, considering six leader ABSs and eight follower USVs, labeled as $k = 1, 2, \cdots, 6$ and $i = 7, 8, \cdots, 14$, respectively. The network topology is given by Fig. 2.

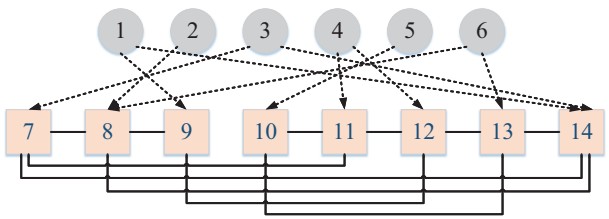

Fig. 2: The communication topology among ABS and USV systems.

Consequently, the matrix $\mathcal{L}_2^{-1}\mathcal{L}_1 \in \mathbb{R}^{8\times 6}$ can be obtained as follows, with each row summing to one.

$$
\mathcal{L}_2^{-1}\mathcal{L}_1 = \begin{bmatrix}
0.1343 & 0.1290 & 0.4124 & 0.1398 & 0.0488 & 0.1356 \\
0.1703 & 0.2852 & 0.1663 & 0.0698 & 0.0232 & 0.2851 \\
0.4324 & 0.1906 & 0.0785 & 0.1369 & 0.0270 & 0.1346 \\
0.0673 & 0.0809 & 0.0892 & 0.1894 & 0.4204 & 0.1528 \\
0.0821 & 0.1241 & 0.1427 & 0.4176 & 0.1317 & 0.1018 \\
0.1269 & 0.2867 & 0.0692 & 0.3410 & 0.0577 & 0.1186 \\
0.1198 & 0.1185 & 0.1248 & 0.1506 & 0.1296 & 0.3567 \\
0.2849 & 0.1065 & 0.3407 & 0.0720 & 0.0403 & 0.1555
\end{bmatrix}.
$$

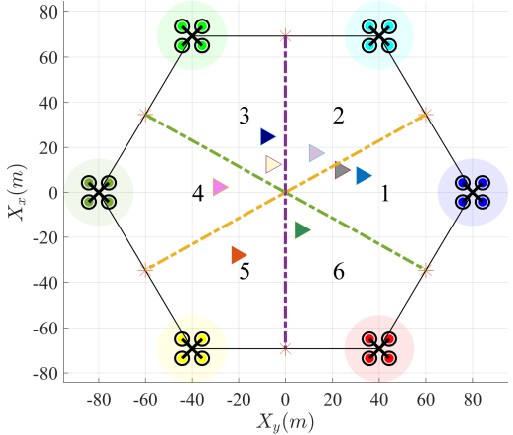

Fig. 3: Regions allocation in two dimensions.

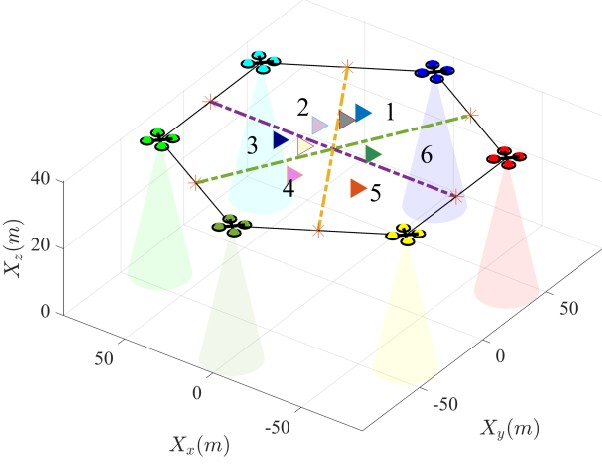

Fig. 4: Regions allocation in three dimensions.

In this case, the allocation index of ABSs corresponding to each USV can be obtained as $\eta = [3, 1, 1, 5, 4, 3, 6, 2]$. Suppose leader ABSs have formed a regular hexagon and $u_k = [0; 0; 0]$. The expected time-varying formation pattern for followers is designed as an ellipse, expressed as $h_7 = h_8 = h_{10} = h_{11} = h_{13} = h_{14} = [15\sin(0.2t); 10\cos(0.2t); \text{atan2}(\dot{h}_{iy}, \dot{h}_{ix}); \dot{h}_{ix}; \dot{h}_{iy}; \dot{h}_{i\phi}]$. $h_{9x} = h_{12x} = 15\sin(0.2t + \pi/2)$ and $h_{9y} = h_{12y} = 10\cos(0.2t + \pi/2)$. USVs share the same ABSs when

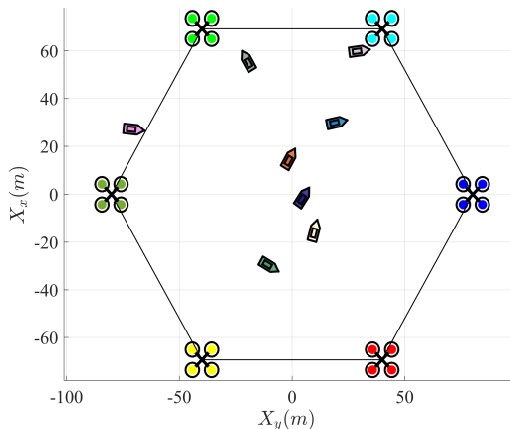

Fig. 5: $t = 0s$ in two dimensions.

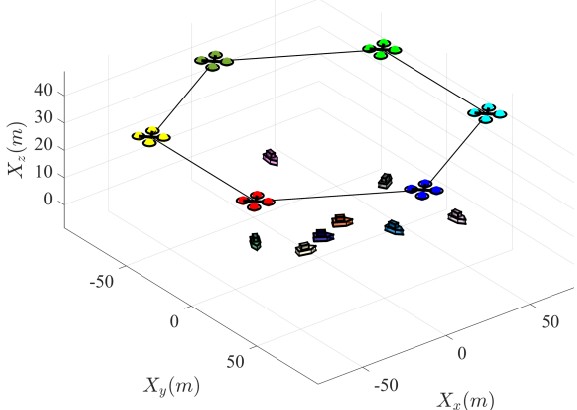

Fig. 6: $t = 0s$ in three dimensions.

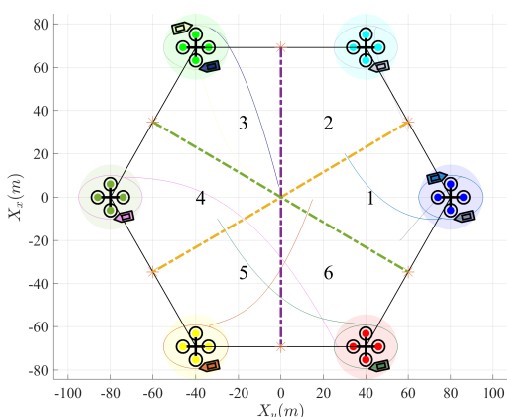

Fig. 7: $t = 60s$ in two dimensions.

$i = 7, 12$ and $i = 8, 9$. The model parameters for the USVs used in this paper are identical to those in reference [14].

The controller parameters are chosen as $\gamma = 10, \alpha_1 = 1, \alpha_2 = 4$ and $K_2 = [-2\boldsymbol{I}_3, -3\boldsymbol{I}_3]$. The feedback matrix can be derived as $K_1 = [-42.043\boldsymbol{I}_3, -16.2193\boldsymbol{I}_3]$. The fuzzy membership functions are selected as $\varphi_{il} = \exp(-\frac{(P_i(l) - o(g))^2}{2})$, where $l = 1, 2, \cdots, 6$ represent the

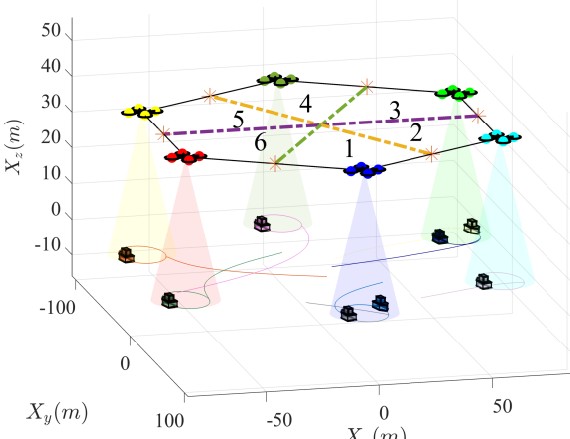

Fig. 8: $t = 60s$ in three dimensions.

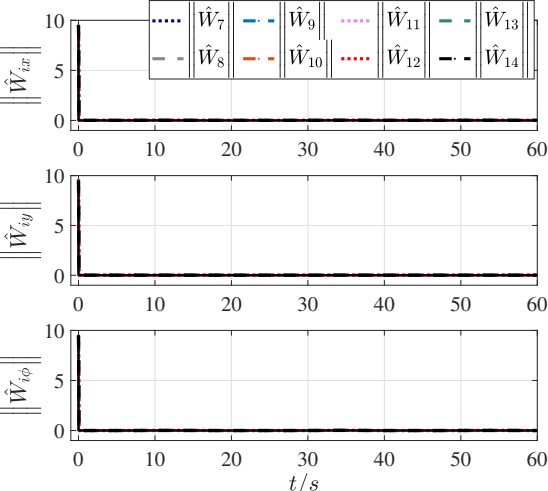

Fig. 9: Adaptive fuzzy parameters in surge, sway, and yaw directions.

current dimension of the vector $P_i$. $o$ ranges from $[-5, 5]$ denote the fuzzy rules and $g = 1, 2, \cdots, 20$ signify the current rule. The fuzzy adaptive update parameters and the initial fuzzy parameter matrix are selected as $\varpi_i = 1$, $\rho_i = 5$, and $W_i(0) = 0$.

The states of ABSs are defined as $P_k = [x_k; y_k; 20; 0; 0; 0]$, where $x_k = r\cos[2(k-1)\pi/M]$, $y_k = r\sin[2(k-1)\pi/M]$, and $r = 80m$ represents the radius of the regular hexagon. The initial states of USVs are chosen as $P_7 = [-1; 5; -\pi/7; 0; 0; 0]$, $P_8 = [56; -20; -\pi/2; 0; 0; 0]$, $P_9 = [30; 20; \pi/10; 0; 0; 0]$, $P_{10} = [15; -1; -\pi/6; 0; 0; 0]$, $P_{11} = [27; -70; \pi/5; 0; 0; 0]$, $P_{12} = [-15; 10; -\pi/4; 0; 0; 0]$, $P_{13} = [-30; -10; \pi/3; 0; 0; 0]$, $P_{14} = [60; 30; \pi/8; 0; 0; 0]$.

Fig. 3 - 4 illustrate the initial allocation regions of ABSs, and Fig. 5 - 8 depict the initial positions and trajectories of eight follower USVs at $t = 0s$ and $t = 60s$, respectively. In Fig. 3 - 4, six ABSs form a regular hexagon and the whole regions are divided equally into six parts. Eight tri-

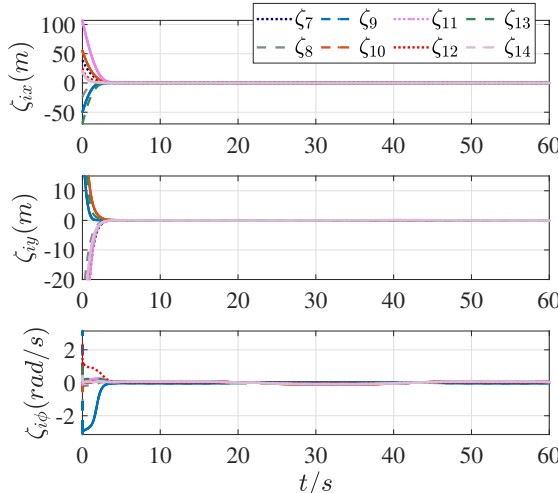

Fig. 10: Curves of time-varying formation errors in surge, sway, and yaw directions.

angles represent the convex states. Perpendicular bisectors are denoted by dot dash lines. In Fig. 7 - 8, all USVs shape the time-varying structures around the allocated ABSs, ensuring they are evenly distributed. The fuzzy adaptive parameters $\hat{W}_i$ are shown in Fig. 9, and the formation errors $\zeta_i$ are given in Fig. 10, where it can be observed that the parameters converge to bounded ranges, and the errors rapidly converge to neighborhoods around the origin. From Fig. 3 - 10, it can be concluded that, all follower USVs successfully form the desired time-varying elliptical formations and rotate around the assigned ABSs, thereby achieving the efficient allocation of a limited number of base stations for each USV.

## V. CONCLUSIONS

The selection of formation centers for the fuzzy time-varying formation problem of follower USVs has been studied in this paper. To determine the allocated leader ABSs for each USV, a Laplacian matrix-based deployment algorithm for ABSs has been designed. In addition, a fuzzy time-varying formation control protocol has been developed for nonlinear USVs, ensuring that all USVs have successfully formed the desired formation pattern, with the allocated ABSs acting as the formation centers. Furthermore, the boundedness of formation errors and fuzzy adaptive parameters has been analyzed using Lyapunov theory. Future research will focus on addressing communication resource constraints in the cooperative control of ABS-USV systems.

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
