# OpenReview forum: "Fuzzy Time-Varying Formation Control for Unmanned Surface Vehicles Considering Aerial Base Station Allocation Algorithm"
_IEEE.org/ICIST/2024/Conference — IEEE ICIST 2024 Conference Submission_

### Official Review · Reviewer_vfPn · 2024-08-29
**This paper can be accepted.**

**Rating:** 10
**Confidence:** 5

**Review:**

This paper focuses on selecting formation centers issue for the time-varying formation control of nonlinear unmanned surface vehicles (USVs) assisted by aerial base stations (ABSs). , A kind of communication volume-based and position-dependent algorithm is designed to obtain the order of the allocated ABSs. An elegant fuzzy time-varying formation control protocol for the USV is proposed. A numerical study is given to verify the effectiveness of the theoretical results.

Overall, this paper is well written and organized. The results contain solid contributions. I suggest that the contributions should be highlighted and summarized in the introduction section.

---

### Official Review · Reviewer_KYzV · 2024-08-30
**Review on Fuzzy Time-Varying Formation Control for Unmanned Surface Vehicles Considering Aerial Base Station Allocation Algorithm**

**Rating:** 7
**Confidence:** 3

**Review:**

1. Is  Assumption 1 reasonable?
2. According to Fig. 4, why don't adaptive parameters converge?
3. What role does region allocation algorithm play in the formation control?

---

### Official Review · Reviewer_WsCB · 2024-08-30
**comment**

**Rating:** 7
**Confidence:** 5

**Review:**

This paper focuses on time-varying formation control of nonlinear unmanned surface vehicles (USVs) assisted by aerial base stations (ABSs). In the reviewer’s opinion, there are some comments which should be addressed by the authors:

1.In the introduction, there are few references and a lack of comparison between this article and the references, which makes it difficult to highlight the contribution of this article; In addition, the innovative content of this article needs to be supplemented.
2.There is only one simulation example in the article and no comparison with existing research methods. It is recommended to add examples or conduct comparisons to improve the validation of the effectiveness of the method proposed in this article.
3.Figures in the simulation are difficult to review due to their small size. Please modify them accordingly.

---

### Decision · Program_Chairs · 2024-09-06

Accept (Oral)